# Implementing quality improvement intervention to improve intrapartum fetal heart rate monitoring during COVID-19 pandemic- observational study

**Pratiksha Bhattarai[1], Rejina Gurung[1,2], Omkar Basnet[1], Honey Malla[1], Mats Målqvist[2‡], Ashish K. C.[2,3‡]***

**1** Golden Community, Jawagal, Lalitpur, Nepal, **2** Department of Women's and Children's Health, Uppsala Global Health Research on Implementation and Sustainability (UGHRIS), Uppsala University, Uppsala, Sweden, **3** Society of Public Health Physicians Nepal, Kathmandu, Nepal

‡ These authors are joint senior authors on this work.
* ashish.k.c@kbh.uu.se

**Data Availability Statement:** All relevant data are within the paper and its Supporting Information files.

## Abstract

### Introduction

Adherence to intrapartum fetal heart rate monitoring (FHRM) for early decision making in high-risk pregnancies remains a global health challenge. COVID-19 has led to disruption of routine intrapartum care in all income settings. This study aims to evaluate the implementation of quality improvement (QI) intervention to improve intrapartum FHRM and birth outcome before and during pandemic.

### Method and materials

We conducted an observational study among 10,715 pregnant women in a hospital of Nepal, over 25 months. The hospital implemented QI intervention i.e facilitated plan-do-study-act (PDSA) meetings before and during pandemic. We assessed the change in intrapartum FHRM, timely action in high-risk deliveries and fetal outcomes before and during pandemic.

### Results

The number of facilitated PDSA meetings increased from an average of one PDSA meeting every 2 months before pandemic to an average of one PDSA meeting per month during the pandemic. Monitoring and documentation of intrapartum FHRM at an interval of less than 30 minutes increased from 47% during pre-pandemic to 73.3% during the pandemic (p<0.0001). The median time interval from admission to abnormal heart rate detection decreased from 160 minutes to 70 minutes during the pandemic (p = 0.020). The median time interval from abnormal FHR detection to the time of delivery increased from 122 minutes to 177 minutes during the pandemic (p = 0.019). There was a rise in abnormal FHR

**Funding:** This study was funded by the Innovation Norway. The funder had no role in study design, data collection and analysis, decision to publish, or preparation of the manuscript.

**Competing interests:** None

**Abbreviations:** COVID-19, Corona Virus of 2019; FHRM, Fetal Heart Rate Monitoring; PoAHS, Pokhara Academy of Health Sciences; REFINE, Rapid Feedback for Quality Improvement in Neonatal Resuscitation; HBB, Helping Babies Breathe; PDSA, Plan-Do-Study-Act; QI, Quality Improvement; SDG, Sustainable Development Goal; LMIC, Low and Middle Income Countries; LIC, Low Income Countries; WHO, World Health Organization; CS, Caesarean Section; FSB, Fresh Stillbirth; NICU, Neonatal Intensive Care Unit; IRB, Institutional Review Board; SPSS, Statistical Package for the Social Sciences; SD, Standard Deviation; IQR, Inter Quartile Range.

detection during the time of admission (1.8% vs 4.7%; p<0.001) and NICU admissions (2.9% vs 6.5%; p<0.0001) during the pandemic.

## Conclusion

Despite implementation of QI intervention during the pandemic, the constrains in human resource to manage high risk women has led to poorer neonatal outcome. Increasing human resources to manage high risk women will be key to timely action among high-risk women and prevent stillbirth.

## Introduction

Globally, 295,000 maternal deaths, 2.0 million stillbirths and 2.5 million newborns deaths occur every year [1–3]. Of these, approximately 94% of the maternal mortality, 85% of the stillbirths and 98% of the neonatal deaths occur in low-and-middle income countries (LMIC) [1–3]. Intrapartum complications account for more than three-fourth of maternal deaths, almost half of the stillbirths and one-fourth of the neonatal mortality [4,5]. High quality care during labor and childbirth is a key to avert these deaths and reduce the burden of fresh stillbirths and early neonatal deaths resulting from birth asphyxia [6–8]. Therefore, intrapartum fetal heart rate monitoring (FHRM) is a key intervention to measure fetal condition during childbirth [9–11].

During the COVID-19 pandemic, disruptions in quality care during labor and childbirth has increased the risks for adverse maternal and fetal outcomes [12,13]. Fear of contracting the disease, diversion of delivery room staffs towards COVID-19 related care, implementation of heterogeneous and inconsistent guidelines in labor and delivery room management have further widened the gap for quality intrapartum care [14,15]. As such, breach in the provision of services like intrapartum FHRM, timely clinical decision making, and effective on-time intervention based on the clinical status of mother and fetus can easily be speculated [16,17].

According to the World Health Organization (WHO), FHR should be monitored every 30 minutes during the first stage of labor, and at least every 15 minutes during the second stage of labor [18–20].

Nepal has made a significant progress in maternal and newborn health in last two decades with reduction in maternal mortality, still birth rate and neonatal mortality rate by 76%, 58% and 68% respectively [21]. However, such substantial gains are at risk due to COVID- 19 pandemic, as our previous study reports that stillbirths and neonatal mortality increased by two and three folds respectively during the initial few months of the pandemic in Nepal [22]. Also, a resilient system in place can be helpful in execution of services like FHR monitoring and undertaking interventions like intrauterine resuscitation, instrumental delivery or a caesarean section timely, without any disruption despite facing various hurdles during the pandemic [23,24].

This study aims to assess the implementation of quality improvement (QI) intervention to improve FHRM and birth outcome before and during pandemic.

## Materials and method

### Study design

This is an observational study to evaluate the QI intervention in the hospital [25]. This study was conducted over a period of 25 months, comparing the period of 13 months before the pandemic (1st March 2019 – 30th March 2020) with the period of 12 months during the pandemic (April 2020 to 31st March 2021).

### Study setting

This study was conducted at Pokhara Academy of Health Sciences (PoAHS), a referral hospital located in Pokhara, Province 4. The hospital provides Comprehensive Emergency Obstetric and Neonatal Care services. The vaginal and instrumental births were conducted in delivery units and the Caesarean births in the operation theatre. The hospital has 500 beds with approximately 7,000 annual deliveries; 15% by Cesarean Section. More specifically, there are 15 admission/waiting, 7 labor and 3 delivery beds, and a team of 7 obstetricians, 5 medical officers and 17 nurse-midwives along with varying number of rotating interns, skilled birth attendants and nursing students (Table 1). Nurses have three duty shifts per day with an average of three nurses and one medical attendant per shift. During the COVID-19 pandemic, these health workers were also assigned for COVID-19 focused care at hospital isolation ward in rotation. After completing 2-week duty in COVID-19 ward, the health workers stayed in quarantine for another two weeks and then resumed their duties at the maternity ward.

In admission room, a midwife thoroughly examines the pregnant women, measures vital signs, FHR, and performs vaginal examination. Thereafter, medical history, vital signs and FHR taken are entered in the confinement book of the labor ward. Doctors on round review the patients chart and provide the initial and subsequent obstetric examination until delivery. After a normal vaginal delivery, mothers and babies are observed in the hospital for 10 to 24 hours in cases with normal delivery. Babies having any complications or requiring medical attention are admitted to the neonatal unit of the hospital.

### Quality improvement intervention

At PoAHS, QI intervention was first introduced to the hospital management committee in January 2019. Brief orientation on QI intervention was provided to hospital leaders and head of departments. Based on the discussions among hospital leaders and management committee, plan to implement QI intervention was developed. From March 2019, QI intervention was implemented at PoAHS. Hospital leaders appointed hospital managers to lead and introduce QI process and facilitate delivery room staffs. The QI intervention consisted of, (a) introducing the use of innovative technologies for fetal heart rate monitoring (Moyo) and (b) Bi-weekly plan- do- study- act (PDSA) meeting [26]. A dashboard was developed to monitor the key quality metrics of maternal and newborn quality care indicators and outcome was established. These indicators were used by health workers to carry out their regular QI (PDSA) meetings (Fig 1).

**Table 1. Number of midwives in the maternity ward before and during pandemic.**

| Duty Shifts | Before Pandemic | During Pandemic |
|---|---|---|
| Morning shift | 4–5 | 3–4 |
| Evening shift | 2–3 | 1–2 |
| Night shift | 3 | 2 |

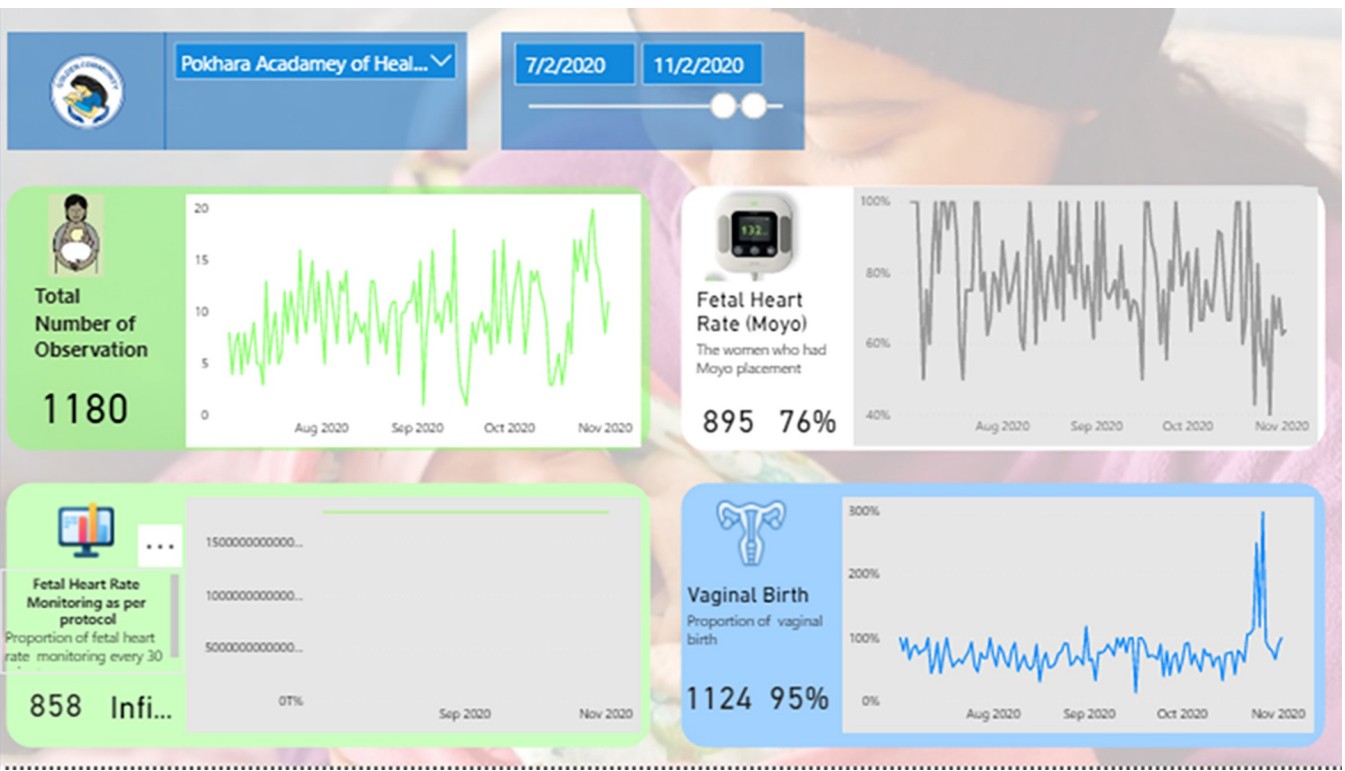

**Fig 1. Dashboard developed to maintain key quality metrics.**

### Implementation of QI intervention before and during COVID-19 pandemic

COVID-19 pandemic severed the hospital and patient management. Providing quality care was challenging to the health workers, however, despite limited human resources and fear of contracting the virus, the midwives continued their routine activities. Continuous communication with hospital leaders and maternity ward in-charge was done by QI coordinator to motivate and facilitate delivery room staffs to continue QI process. Head of department and ward in-charge along with QI coordinator encouraged midwives to conduct their regular PDSA meetings, use of Moyo for FHR monitoring, and practice daily skill drills to strengthen quality care even during the pandemic following the infection prevention protocol. Firstly, to deliver an improved quality intrapartum care, midwives conducted regular PDSA meetings where they identified their problems, prioritized it and planned a possible solution. Out of total 21 PDSA meetings conducted over a period of 25 months, 13 PDSA meetings were conducted during the period of pandemic where midwives continued to discuss on improving the key quality care indicator, fetal heart rate monitoring and documentation and constantly worked to achieve their target. Secondly, to prevent COVID-19 transmission, midwives also oriented mothers to identify normal and abnormal FHR detected by Moyo and to inform them if any abnormality occurred. Midwives provided required intervention responding to mother's condition (Table 2).

### Study population

The study participants included mothers in labor with an estimated gestational age more than 22 weeks and birth weight more than 500 grams. Women who consented to get enrolled were

**Table 2. Brief discussion note during each PDSA meeting.**

| PDSA Meeting | Problems Identified | Action Taken |
|---|---|---|
| 1st PDSA– 8th PDSA | Bag and Mask ventilation not initiated within Golden 1 minute. Babies being deprived of the benefits of Delayed cord clamping. | Pre- assemble necessary equipment. Take minimal time while stimulating and suctioning. Replacing blunt episiotomy scissors. Reminding the nursing staffs about the importance of delayed cord clamping. Reinforcing the staffs to do delay cord clamp. Nurses coordinated with nursing students and initiate resuscitation while baby is on the mother's abdomen with cord intact. |
| 9th PDSA– 12th PDSA | Reduced FHR monitoring Bag and Mask ventilation not initiated within Golden 1 minute. | Using Moyo equipment for FHR monitoring. Ward In-charge initiated staffs in monitoring FHR until delivery. Staffs will monitor FHR immediately after admission. |
| 13th PDSA- 16th PDSA | Breast-Feeding not initiated within 1 hour after birth. Bag and Mask ventilation not initiated within Golden 1 minute. | Encourage family members to initiate early breast feeding. Female attendant be allowed in labour room for helping mothers in breast feeding. Awareness about feeding to mothers and visitors. Breast feeding within 1 hour of birth should be done. |
| 17th PDSA– 20th PDSA | Skin to skin contact not initiated immediately after birth. Bag and Mask ventilation not initiated within Golden 1 minute. | Initiating skin to skin contact in every crying babies. |

included in the study. Exclusions included multiple pregnancies, critically ill patients and cases with undetectable FHR or whose FHR was absent on admission.

## Data collection

For this study, a validated clinical observation checklist was used to observe the labor and delivery event for all vaginal births, and women's obstetric and neonatal information was collected from patient charts and case notes. A data collection system was set up at hospital and observations were done by independent clinical research officer.

## Data management and analysis

All the data entered in the tablet based application were reviewed on a weekly basis by an independent data base manager and discussions and clarifications were done with surveillance officers after reviewing the entered data. For this study, data were extracted into SPSS software (IBM SPSS Statistics for Windows, Version 23.0) for the cleaning of the extracted data of all births and observed data of all deliveries occurring in the hospital. Data consistency was checked and mismatched cases were retrieved and corrected accordingly before the analysis.

Mean (SD), Median (IQR) and proportions were used for descriptive analysis of the background variables and pre–pandemic and during the pandemic maternal and neonatal outcomes. Pearson's Chi- square tests were used to test for proportion differences. Mann-Whitney U test and independent t-test were used to compare group median and mean respectively. The use of Mann Whitney U or independent t-test for a continuous variable was based on the normality of the distribution using the histogram on skewness.

## Variables

**Primary outcome.** The primary outcome measure was FHR defined as normal (100–160 beats per minute) and abnormal (absent, <100 or >160 bpm).

**Secondary outcome.** The secondary outcome included the APGAR score at 5 minutes (abnormal was defined as APGAR Score <7); mode of delivery (vaginal delivery, CS, instrumental), perinatal outcome at birth [i.e. normal, admission to the neonatal unit, intrapartum stillbirth (defined as those babies having FHR present during the intrapartum period and 15 minutes before birth, but were born without having any signs of life), and neonatal outcome at 24 hours [i.e. normal, referred to higher center or transferred to Neonatal Intensive Care Unit (NICU), FSB, First day Mortality (babies dying within 24 hours after birth)].

**Socio-demographic characteristics.** For sociodemographic characteristics, we included maternal age (<20, 20–35 and >35 years), parity defined as women who hasn't given birth to a child previously (nulliparous), women who has given birth once or carried a pregnancy beyond 28 weeks previously (primiparous) and women having born more than one child previously (multiparous).

**Obstetric characteristics.** Obstetric Characteristics included complications during the time of admission (includes pre-eclampsia, eclampsia, diabetes, fever, premature rupture of membrane, pre-term premature rupture of membrane, polyhydramnios, oligohydramnios, cephalo-pelvic disproportion, breech / transverse lie, prolonged Labor, decrease Fetal movements, ante-partum hemorrhage, chorioamnionitis, cord Prolapse and cord around the neck), mode of delivery which included vaginal delivery, instrumental delivery and Caesarean births) and mothers having complications during or at the time of delivering the baby.

Time intervals included admission to first abnormal FHR detection, first abnormal heart rate detection to time of delivery.

**Neonatal characteristics.** For neonatal characteristics, we included preterm births (defined as <37 weeks of gestation on the basis of first day of mothers last menstrual period) and low birth weight babies (defined as baby's birth weight ≤ 2500 grams).

**Ethics statement.** For this study, ethical approval (no. 87/2018) was received from the national ethical review board, Nepal Health Research Council (NHRC). Ethical clearance was also received from the Institutional Review board (IRB) from the hospital where the study was conducted. Written consent was taken from the pregnant women who agreed to participate before the extraction and clinical observations.

## Result

During the study period, 15,797 women were admitted in hospital for delivery and 15,184 (96.1%) were eligible for enrolment, of whom 14,584 (92.3%) consented to participate in the study. Women whose FHR was absent at the time of admission were excluded from the study. Of the total 14,584 participants, 10,715 women were observed for their delivery events of which 7,459 (69.6%) women participated before the pandemic and 3,256 (30.4%) women participated during the pandemic (Fig 2).

The figure presents the monthly change in intrapartum FHRM practice and PDSA Meetings conducted before and during the pandemic. On an average the number of facilitated PDSA meetings increased from an average of one PDSA meeting at an interval of two months to an average of one PDSA meeting per month during the period of pandemic. During the pre-pandemic period, the intrapartum FHRM ranged from 1% to the maximum of 11% while it ranged from 7% to the maximum of 66% during the pandemic (Fig 3).

The mean age of women during the pre-pandemic period was 24.5 years (SD 4.65) which increased to 25.8 (SD 4.74) during the pandemic (p<0.0001). There was an increase in delivery by nulliparous women from 1.0% births pre- pandemic to 9.2% births during pandemic (p<0.0001). The proportion of women who had a complication during admission increased from 5.5% pre pandemic to 12.5% during the pandemic (p <0.0001). The proportion of babies

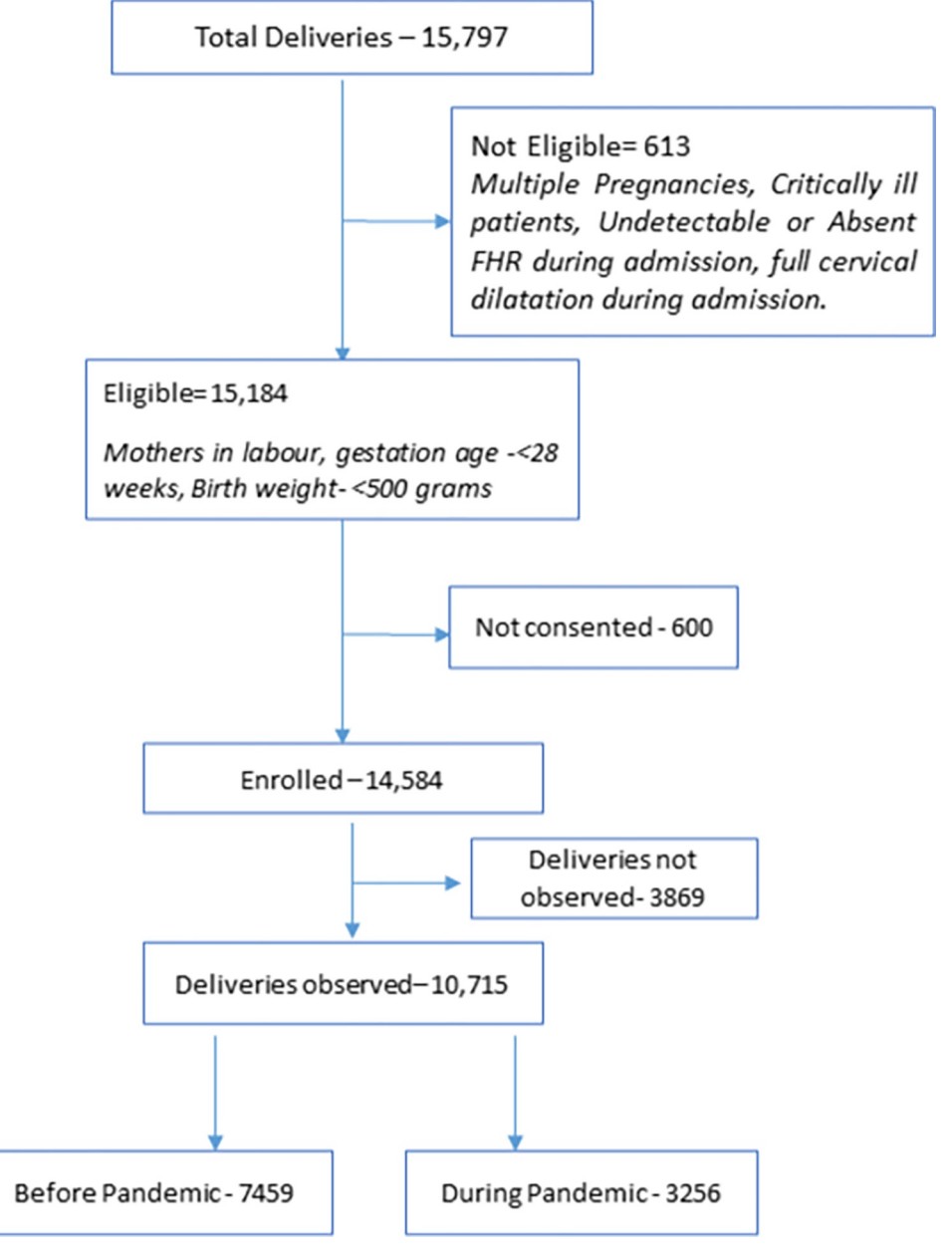

**Fig 2. Flow chart showing the study participants.**

born preterm (before 37 weeks) increased from 4.6% before pandemic to 6.4% during the pandemic; (p<0.0001) and there was an increased proportion of the babies born low birth weight during the pandemic (8.1% vs 10.2%, p<0.0001) (Table 3).

Overall, the frequency of intrapartum FHR monitoring when compared to pre-pandemic period increased during the pandemic (p<0.0001). Monitoring and documentation of intrapartum FHR in an interval of less than 30 minutes increased from 47% in pre-pandemic period to 73.3% during the pandemic (p<0.0001). Among women whose intrapartum FHR was detected abnormal during labor, an increased proportion of women had their FHR monitored as per protocol during pandemic (when compared with pre-pandemic period (59.3% vs 20.0%,

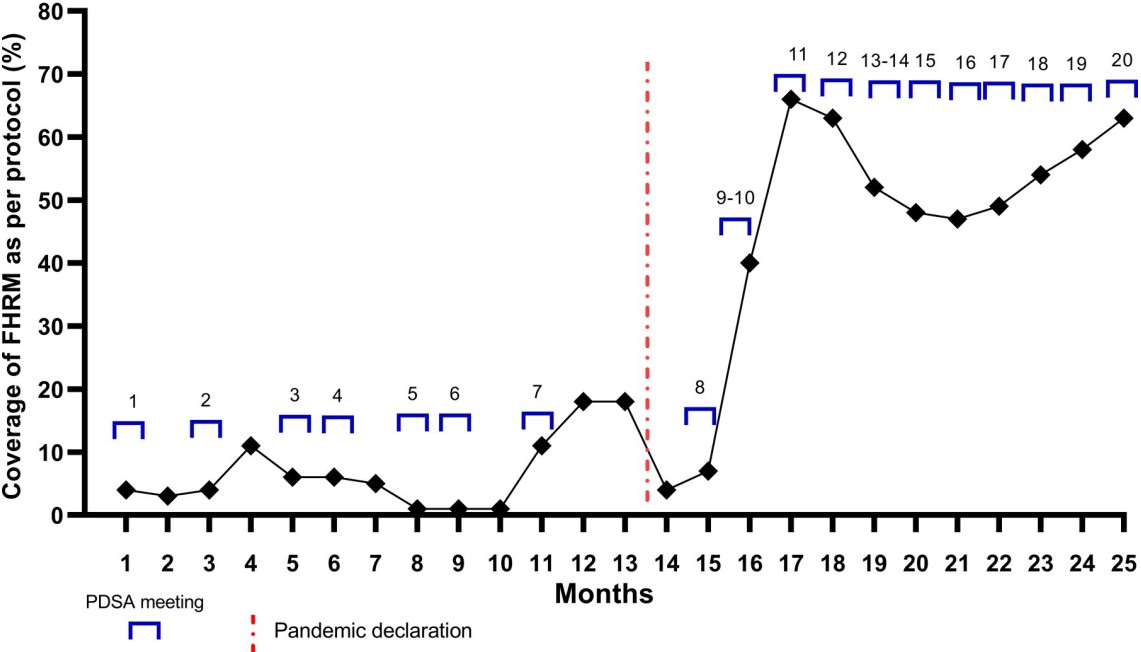

**Fig 3. Monthly FHRM practice before and during pandemic.**

p<0.0001). The median time interval from admission to abnormal heart rate detection decreased from 160 minutes in pre pandemic period to 70 minutes during the pandemic (p = 0.020). The median time interval from abnormal FHR assessment to delivery increased from 122 minutes at pre-pandemic period to 177 minutes during the pandemic (p = 0.019). The median time duration of continuous Moyo application was 173-minutes pre- pandemic versus 148-minutes during the pandemic (p = 0.132), respectively (Table 4).

When compared with pre-pandemic period, during the pandemic an increased proportion of women was detected with abnormal fetal heart rate at the time of labor (1.8% vs 4.7%); (p<0.0001). Overall, the proportion of newborns with APGAR score <7 at 5 minutes increased from 2.7% before the pandemic to 4.5% during the pandemic; (p<0.0001). There was an increase in proportion of babies who did not cry immediately after birth during the pandemic when compared with pre-pandemic period (4.5% vs. 7.7%; p<0.0001). There was an increase in proportion of newborns receiving bag and mask ventilation (15.9% vs. 26.3%) during the pandemic as compared to pre-pandemic period. Increased proportion of babies were admitted to a neonatal intensive care unit (2.9% vs 6.5%; p <0.0001) immediately after the birth and fresh stillbirth (0.7% vs 0.9%; p<0.0001) during the period of pandemic when compared with pre-pandemic period (Table 5).

## Discussion

Intrapartum FHR monitoring and documentation done by health workers in the study site increased during the pandemic. Frequency of intrapartum FHR monitoring and recording in an interval of both less than 30 minutes and greater than 30 minutes increased during the pandemic. In our study, we observed increase in the proportion of women presenting with complications at the time of admission during the time of pandemic. The median time from admission to abnormal heart rate detection decreased by 90 minutes.

**Table 3. Demographic and obstetric characteristics of women delivering in the hospital.**

| Category | Pre- Pandemic (N = 7,459) | During- Pandemic (N = 3,256) | Total (N = 10,715) | P-Value |
|---|---|---|---|---|
| AGE | 7459 | 3256 | 10715 | <0.0001 |
| Mean±SD | 24.54±4.65 | 24.79±4.74 | 24.61±4.68 | |
| < 20 | 550(7.4%) | 186(5.7%) | 736(6.9%) | |
| 20–35 | 6773(90.8%) | 2982(91.6%) | 9755(91.0%) | |
| >35 | 136 (1.80%) | 88(2.70%) | 224(2.10%) | |
| Parity* | 5150 | 3252 | 8402 | <0.0001 |
| Nulli Para | 54(1.0%) | 298(9.2%) | 352(4.2%) | |
| Primi Para | 2792 (54.2%) | 1472(45.3%) | 4264(50.7%) | |
| Multi Para | 2304(44.7%) | 1482(45.6%) | 3786(45.1%) | |
| Gestational Age* | 7071 | 3256 | 10327 | <0.0001 |
| Mean±SD | 39.11±1.67 | 39.01±1.98 | 39.08±1.77 | |
| Preterm | 328(4.6%) | 208(6.4%) | 536(5.2%) | |
| Term | 6743(95.4%) | 3048(93.6%) | 9791(94.7%) | |
| Admission Complication* | 3479 | 3252 | 6731 | <0.0001 |
| No | 3288(94.5%) | 2845(87.5%) | 6133(91.1%) | |
| Yes | 191(5.5%) | 407(12.5%) | 598(8.9%) | |
| Birth Weight* | 7324 | 3249 | 10573 | <0.0001 |
| Mean±SD | 3037.23±461.62 | 2990.09±482.58 | 3022.74±468.65 | |
| <2500 | 591(8.1%) | 333(10.2%) | 924(8.7%) | |
| 2500–3500 | 5900 (80.6%) | 2565 (78.9%) | 8465(80.1%) | |
| >3500 | 833 (11.4%) | 351(10.8%) | 1184(11.2%) | |

*Variables with missing information.

Data shown as n (%) unless otherwise stated. SD: Standard Deviation.

**Table 4. Frequency of intrapartum FHR monitoring pre and during pandemic period.**

| Category | Pre- Pandemic | During- Pandemic | P-Value |
|---|---|---|---|
| Fetal Heart Rate Monitoring | 7459 | 3256 | <0.0001 |
| < 30 Minutes | 3503 (47.0%) | 2387(73.3%) | |
| >30 Minutes | 1633 (21.9%) | 811 (24.9%) | |
| FHR not monitored | 2323(31.1%) | 58(1.8%) | |
| FHRM practice in mothers with Abnormal Heart rate detection | Pre-Pandemic | During-Pandemic | P- Value |
| | n = 123* | n = 154 | <0.0001 |
| As per protocol | 23 (18.7%) | 83 (59.3%) | |
| Sporadically | 95 (77.2%) | 58 (37.7%) | |
| Yes, only once | 5 (4.1%) | 13 (8.4%) | |
| Time intervals (q1, q3)* | Pre-Pandemic | During-Pandemic | P- Value |
| | n = 123* | n = 154 | 0.020 |
| Time from admission to Abnormal FHR detection median (Q1, Q3) in minutes | 160(36,300) | 70(10,297) | |
| | n = 123* | n = 154 | 0.019 |
| Time from detection of abnormal FHR to delivery median (Q1, Q3) in minutes | 122(53,249) | 177(79,309) | |
| | n = 41 | n = 121 | 0.132 |
| Duration of Continuous Moyo median (Q1, Q3) in minutes | 173(108,313) | 148(65,250) | |

*Variables with missing information.

**Table 5. Comparison of perinatal and neonatal outcome pre-pandemic and during the time of pandemic.**

| Variable | Pre- Pandemic | During- Pandemic | Total | P-Value |
|---|---|---|---|---|
| FHR monitoring during labor | **7459** | **3256** | **10715** | **<0.0001** |
| No | 2323(31.1%) | 58(1.8%) | 2381(22.2%) | |
| Yes | 5136(68.9%) | 3198(99.2%) | 8334 | |
| FHR during labor | **7459** | **3256** | **10715** | **<0.0001** |
| Normal | 7324(98.2%) | 3102(95.3%) | 10426(97.3%) | |
| Abnormal | 135(1.8%) | 154(4.7%) | 289(2.7%) | |
| Mode of delivery | **7459** | **2978** | **10715** | **<0.0001** |
| Normal | 6478 (86.8%) | 2978 (91.5%) | 9456 (88.2%) | |
| Instrumental | 188(2.5%) | 97 (3.0%) | 285(2.7%) | |
| CS | 793(10.6%) | 181(5.6%) | 974(9.1%) | |
| Complications to mother at the time of delivery* | **5035** | **3231** | **8266** | **<0.0001** |
| No | 4997 (99.2%) | 3174 (98.2%) | 8171(98.9%) | |
| Yes | 38(0.8%) | 57(1.8%) | 95(1.1%) | |
| Apgar Score at 5* | **5105** | **3253** | **8072** | **<0.0001** |
| >7 | 4965(97.3%) | 3107(95.5%) | 8072(96.6%) | |
| <7 | 140(2.7%) | 146(4.5%) | 286(3.4%) | |
| Crying at birth* | **6666** | **3075** | **9741** | **<0.0001** |
| No | 301(4.5%) | 236(7.7%) | 537(5.5%) | |
| Yes | 6365(95.5%) | 2839(92.3%) | 9204(94.5%) | |
| Bag and Mask Ventilation | **301** | **236** | **537** | **0.003** |
| No | 253(84.1%) | 174(73.7%) | 427(79.5%) | |
| Yes | 48(15.9%) | 62(26.3%) | 110(20.5%) | |
| Delivery Outcome* | **6666** | **3075** | **9741** | **<0.0001** |
| Live birth | 6420(96.3%) | 2844(92.5%) | 9264(95.1%) | |
| Transfer to NICU | 194(2.9%) | 201(6.5%) | 395(4.1%) | |
| Fresh Stillbirth | 49(0.7%) | 27(0.9%) | 76(0.8%) | |
| First day Mortality | 3(0.0%) | 3(0.1%) | 6(0.1%) | |

*Variables with missing information.

There are several potential reasons for increase in detection of abnormal intrapartum FHR at admission. Firstly, the number of complicated cases increased due to the lockdown restrictions and extreme fear of COVID-19 infection resulting in three delays; delay in seeking care, reaching health facility and receiving care from health care providers. This has highlighted on more women seeking health facility care only after complications has arisen. Secondly, more complicated cases arriving health facility may have resulted in more FHR abnormalities and subsequent rise in abnormal heart rate detection. Third, there is lack of dedicate human resource to manage high risk pregnancy early on.

Despite improvement in intrapartum fetal heart rate monitoring and early abnormal heart rate detection, there was delay in providing timely and appropriate interventions to both mothers and their newborns, median time from first abnormal heart rate detection to time of delivery increased by nearly an hour. We observed newborns who did not cried immediately after birth and babies receiving resuscitation increased during pandemic nearly by 2-fold. Babies requiring intensive care or transferred to other health care facility for NICU admission increased nearly by 3-fold along with increase in intrapartum stillbirth and first day mortality. Health workers working in limited resources, lack of human resources, diversion of health

workers for COVID-19 focused care, and lack of timely decision making might have disrupted the proper service delivery by health care system [27,28].

Ever since COVID-19 pandemic has strained the health care system, institutional maternal mortality rate, stillbirth and neonatal mortality have been increasingly high [12]. In low resource settings like Nepal, factors contributing to these maternal deaths, stillbirths and neonatal deaths in health facility can bring detrimental effects in achieving country's ambitious Sustainable Development Goal (SDG) of reducing stillbirths and newborn deaths by 2030 [29].

In this study, we observed that implementation of QI intervention comprising PDSA Meetings contributed to improve health worker's performance in intrapartum FHRM. However, rise in adverse fetal outcome in health facility point towards lack in clinical decision making and failure of undertaking appropriate intervention for maternal and newborn survival which raise questions to the number of skilled health workers in health facility. Nepal is found to have 0.67 doctors and nurses per 1,000 populations which is against WHO recommendation of 2.3 doctors and nurses per 1,000 populations [30]. Technologies can guide health worker for better decision making, but human resource working on it can only bring desirable changes only if they are adequate in number. The current health workers need to be increased four-fold to optimize the service for timely intervention for women who had abnormal FHR during labor. In Nepal, government hospitals are always overloaded with cases where majority of the population seek health care; technologies can support these workforces only to some extent; however, timely and effective intervention solely depends upon obstetric workforce. It's now high time that government should prompt their priorities in maintaining doctors, nurses and patient's ratio in an effort to provide better service and care to women and their newborns.

## Methodological consideration

Observational data collection and documenting trends in intrapartum FHR monitoring practices by trained surveillance officers in labor room despite the fear of contracting the disease in a large group of mothers makes this a rare and important study. However, we have some limitations. Our single-centered study lacks the information on health workers practice of FHR monitoring in COVID-19 positive mothers and the associated fetal outcomes as we don't have information on the COVID-19 status of the mothers. Owing to the workload inside the delivery room, observer bias can't be ruled out for observation of health worker's performance.

## Conclusion

Despite the improvement in intrapartum FHR monitoring and documentation, adverse intrapartum related events and neonatal deaths increased during Covid-19 pandemic. Detection of FHR abnormalities should be followed by early clinical decision making and undertaking of appropriate interventions timely for better fetal outcomes. While the implementation of quality improvement intervention can make intrapartum care better, strengthening health system resilience is vital to prevent stillbirths and neonatal deaths in low resource settings. Appraisal of health worker's performance on quality care in labor and delivery room through regular monitoring and supervision should be a priority of the government to capacitate health workforce that can effectively execute maternal and newborn services amidst any crisis like pandemics.

## Supporting information

**S1 Dataset.**
(CSV)

**S1 File.**
(CSV)

## Acknowledgments

### Declaration

We want to acknowledge and thank all the data collectors for the data collection and management.

### Ethical consideration

For this study, ethical approval (no. 87/2018) was received from the national ethical review board, Nepal Health Research Council (NHRC). Ethical clearance was also received from the Institutional Review board (IRB) from the hospital where the study was conducted. Written consent was taken from the pregnant women who agreed to participate before the extraction and clinical observations. Surveillance officers were trained on maintaining the confidentiality of the information related to mothers and their newborns.

## Author Contributions

**Conceptualization:** Pratiksha Bhattarai.

**Formal analysis:** Rejina Gurung, Omkar Basnet, Mats Målqvist.

**Funding acquisition:** Ashish K. C.

**Investigation:** Pratiksha Bhattarai, Rejina Gurung, Honey Malla.

**Methodology:** Omkar Basnet, Ashish K. C.

**Project administration:** Pratiksha Bhattarai, Rejina Gurung, Omkar Basnet, Honey Malla.

**Software:** Omkar Basnet.

**Supervision:** Honey Malla, Mats Målqvist, Ashish K. C.

**Validation:** Ashish K. C.

**Visualization:** Omkar Basnet.

**Writing – original draft:** Pratiksha Bhattarai.

**Writing – review & editing:** Rejina Gurung, Omkar Basnet, Honey Malla, Mats Målqvist, Ashish K. C.

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
