## [Decision Letter · Decision Letter 0]

17 May 2022

PONE-D-21-29474Implementing Quality Improvement Interventions to improve Fetal heart rate monitoring during COVID-19 pandemic- observational studyPLOS ONE

Dear Dr. KC,

Thank you for submitting your manuscript to PLOS ONE. After careful consideration, we feel that it has merit but does not fully meet PLOS ONE’s publication criteria as it currently stands. Therefore, we invite you to submit a revised version of the manuscript that addresses the points raised during the review process.

Please see the reviewer comments below, and address each of the comments in your revised manuscript.

We look forward to receiving your revised manuscript.

Kind regards,

Hugh Cowley

Staff Editor

PLOS ONE

**Journal requirements:**

“This study was funded by the Innovation Norway”

" This study was funded by the Innovation Norway”

“This study was funded by the Innovation Norway”

“None”

Reviewers' comments:

Reviewer's Responses to Questions

**Comments to the Author**

1. Is the manuscript technically sound, and do the data support the conclusions?

Reviewer #1: Yes

2. Has the statistical analysis been performed appropriately and rigorously? 

Reviewer #1: Yes

3. Have the authors made all data underlying the findings in their manuscript fully available?

Reviewer #1: Yes

4. Is the manuscript presented in an intelligible fashion and written in standard English?

Reviewer #1: Yes

5. Review Comments to the Author

Reviewer #1: 1. Why did you use median for time interval analysis? Had its normality been tested? Could you provide an explanation for this (if it's abnormal, then median was used,; if it's normal, then mean should have been used)

2. Was the decreased interval from admission to abnormality detection decreased due to lack of human resources? It should have been presented in the discussion section.

3. Was the increased number of infants admitted to NICU during Covid 19 pandemic because of the good system or the pandemic itself or poor input of the infants? Please present it in the discussion section.

4. The conclusion reports the quality improvement, yet there was lack of human resources. Would you explain how many health workers should the hospital have had?

5. The article has generally been written in the standard English and is relatively easy for the readers to understand its content. There are, however, several or minor mistakes that have to be corrected. The mistakes are contained in both the Abstract and main body of the article. They include grammatical errors (inappropriate sentence structures), inconsistent use of the tenses (the result of a study is commonly written in the Simple Past Tense instead of the Simple Present Tense), and inappropriate use of adverbs and prepositions for some verbs. I hope the author will correct the errors prior to the publication.

6. PLOS authors have the option to publish the peer review history of their article (what does this mean?). If published, this will include your full peer review and any attached files.

Reviewer #1: **Yes: **Ekawaty Lutfia Haksari

---

## [Author Response · Author response to Decision Letter 0]

26 May 2022

24 May 2022

Response to reviewer’s comment

Comment 1#. Why did you use median for time interval analysis? Had its normality been tested? Could you provide an explanation for this (if it's abnormal, then median was used,; if it's normal, then mean should have been used)

Response- The time interval from admission to abnormal fetal heart rate detection and time interval from abnormal FHR detection to time of delivery is not normally distributed. We did a normality test to respond to the reviewer’s query, please find below the time distribution using the histogram. 

Comment 2#. Was the decreased interval from admission to abnormality detection decreased due to lack of human resources? It should have been presented in the discussion section.

Response- There are two possible reasons for decreased interval from admission to abnormality detection. First, is the lack of dedicated human resource to manage the high-risk mothers early on as the proportion of CS decreased as shown in table 5. Second, there is an increase in proportion of women admitted with complication at birth (5.5% before pandemic vs 12.5% during pandemic) as noted in table 3. We have added in the discussion section.

Comment 3#. Was the increased number of infants admitted to NICU during Covid 19 pandemic because of the good system or the pandemic itself or poor input of the infants? Please present it in the discussion section.

Response- The reasons for increase in number of infants admitted to NICU during pandemic despite improvement in early detection of abnormal fetal heart rate are same as above. First, the proportion of complication at admission increased due to increase in proportion of women with obstetric complication increased during pandemic. Second, the time to action despite early detection is lack of human resource to act upon women who require the care most. So, the pandemic has induced all two delays in care, delay in women coming to the health facilities and delay in high quality care.

Comment 4#. The conclusion reports the quality improvement, yet there was lack of human resources. Would you explain how many health workers the hospital should have had?

Response- We have now provided the estimated number of health workers the hospital should have to timely action on women who had abnormal FHR during labour. 

Comment 5#. The article has generally been written in the standard English and is relatively easy for the readers to understand its content. There are, however, several or minor mistakes that have to be corrected. The mistakes are contained in both the Abstract and main body of the article. They include grammatical errors (inappropriate sentence structures), inconsistent use of the tenses (the result of a study is commonly written in the Simple Past Tense instead of the Simple Present Tense), and inappropriate use of adverbs and prepositions for some verbs. I hope the author will correct the errors prior to the publication.

Response- We have now done the copy edit.

---

## [Decision Letter · Decision Letter 1]

26 Sep 2022

Implementing Quality Improvement Interventions to improve Fetal heart rate monitoring during COVID-19 pandemic- observational study

PONE-D-21-29474R1

Dear Dr. KC,

We’re pleased to inform you that your manuscript has been judged scientifically suitable for publication and will be formally accepted for publication once it meets all outstanding technical requirements.

Kind regards,

Linglin Xie

Academic Editor

PLOS ONE

Additional Editor Comments (optional):

Reviewers' comments:

Reviewer's Responses to Questions

**Comments to the Author**

1. If the authors have adequately addressed your comments raised in a previous round of review and you feel that this manuscript is now acceptable for publication, you may indicate that here to bypass the “Comments to the Author” section, enter your conflict of interest statement in the “Confidential to Editor” section, and submit your "Accept" recommendation.

Reviewer #1: All comments have been addressed

2. Is the manuscript technically sound, and do the data support the conclusions?

Reviewer #1: Yes

3. Has the statistical analysis been performed appropriately and rigorously? 

Reviewer #1: Yes

4. Have the authors made all data underlying the findings in their manuscript fully available?

Reviewer #1: Yes

5. Is the manuscript presented in an intelligible fashion and written in standard English?

Reviewer #1: Yes

6. Review Comments to the Author

Reviewer #1: (No Response)

7. PLOS authors have the option to publish the peer review history of their article (what does this mean?). If published, this will include your full peer review and any attached files.

Reviewer #1: No

---

## [Editor Report · Acceptance letter]

2 Oct 2022

PONE-D-21-29474R1 

Implementing Quality Improvement Intervention to improve Intrapartum Fetal heart rate monitoring during COVID-19 pandemic- observational study 

Dear Dr. KC:

I'm pleased to inform you that your manuscript has been deemed suitable for publication in PLOS ONE. Congratulations! Your manuscript is now with our production department. 

Kind regards, 

on behalf of

Dr. Linglin Xie 

Academic Editor

PLOS ONE